# Current Uses and Future Perspectives of Genomic Technologies in Clinical Microbiology

**DOI:** 10.3390/antibiotics12111580

**Published:** 2023-10-30

**Authors:** Irene Bianconi, Richard Aschbacher, Elisabetta Pagani

**Affiliations:** Laboratory of Microbiology and Virology, Provincial Hospital of Bolzano (SABES-ASDAA), Lehrkrankenhaus der Paracelsus Medizinischen Privatuniversitätvia Amba Alagi 5, 39100 Bolzano, Italy; richard.aschbacher@sabes.it (R.A.); elisabetta.pagani@sabes.it (E.P.)

**Keywords:** antimicrobial resistance, microbial genomics, genome sequencing, metagenomics

## Abstract

Recent advancements in sequencing technology and data analytics have led to a transformative era in pathogen detection and typing. These developments not only expedite the process, but also render it more cost-effective. Genomic analyses of infectious diseases are swiftly becoming the standard for pathogen analysis and control. Additionally, national surveillance systems can derive substantial benefits from genomic data, as they offer profound insights into pathogen epidemiology and the emergence of antimicrobial-resistant strains. Antimicrobial resistance (AMR) is a pressing global public health issue. While clinical laboratories have traditionally relied on culture-based antimicrobial susceptibility testing, the integration of genomic data into AMR analysis holds immense promise. Genomic-based AMR data can furnish swift, consistent, and highly accurate predictions of resistance phenotypes for specific strains or populations, all while contributing invaluable insights for surveillance. Moreover, genome sequencing assumes a pivotal role in the investigation of hospital outbreaks. It aids in the identification of infection sources, unveils genetic connections among isolates, and informs strategies for infection control. The One Health initiative, with its focus on the intricate interconnectedness of humans, animals, and the environment, seeks to develop comprehensive approaches for disease surveillance, control, and prevention. When integrated with epidemiological data from surveillance systems, genomic data can forecast the expansion of bacterial populations and species transmissions. Consequently, this provides profound insights into the evolution and genetic relationships of AMR in pathogens, hosts, and the environment.

## 1. Introduction

In recent years, clinical microbiology has undergone a remarkable transformation, thanks to the integration of genomics. Breakthroughs in sequencing technologies, especially over the past decade, have made high-throughput genome sequencing not only cost-effective but also accessible. This technological jump has empowered clinical microbiologists to overcome the limitations of traditional methods.

The COVID-19 pandemic serves as a compelling example of genomics’ pivotal role [1]. Next-generation sequencing (NGS) played a crucial role in tracking genetic variants of the virus, expediting vaccine development and monitoring its global spread.

Genome sequencing platforms have significantly reduced the time required for identifying infectious disease agents and characterising newly emerging pathogens, such as the Zika and Ebola viruses [2,3]. Whole-genome sequencing (WGS) has become essential for tracing the transmission of infectious agents within healthcare settings. By comparing the genomes of isolates from different patients, researchers can pinpoint the sources of outbreaks and trace the routes of transmission [4]. Moreover, NGS technologies have enabled the analysis of complex microbial communities, shedding light on the role of the human microbiome in health and disease. Metagenomics has also emerged as a formidable tool for diagnosing infections caused by fastidious or unculturable organisms and exploring the diversity of microorganisms in various clinical samples [5,6].

NGS has provided clinical microbiologists with powerful tools for detecting and monitoring antimicrobial resistance (AMR) [7,8]. The growing threat of AMR underscores the necessity of integrating genomics into efforts to combat this global health challenge. The genomic surveillance of AMR is vital for comprehending how resistance evolves, predicting effective treatments, and making informed decisions regarding patient care.

This review aims to explore these recent breakthroughs and emerging trends, shedding light on the profound impact of genomics on clinical microbiology. By examining how genomics can be employed in pathogen identification, and understanding transmission, monitoring AMR, and investigating outbreaks, this review seeks to provide a comprehensive overview of genomics’ central role in modern clinical microbiology. It will also consider the challenges and opportunities ahead, with particular emphasis on the One Health approach, which acknowledges the interconnectedness of human, animal, and environmental health. This holistic perspective is essential for comprehending and mitigating emerging infectious diseases.

## 2. Nucleic Acid Sequencing Technologies and Their Evolution

The progress made in sequencing technology, from its launch to the current era, has been extremely remarkable. Next-generation sequencing (NGS) stands out as a revolutionary technique, enabling the parallel and simultaneous sequencing of billions of DNA fragments. NGS applications have evolved beyond their initial research role, becoming diagnostic methods steadily gaining prevalence within clinical microbiology laboratories (see Table 1).

### 2.1. First-Generation Sequencing

In 1975, approximately 22 years after Watson and Crick published their groundbreaking work on the double-helix structure of DNA [9], Sanger and Coulson introduced their pioneering DNA sequencing protocols [10]. Sanger developed a method known as “chain-termination”, which relied on four polymerisation reactions. Each of these reactions involved tritium-radiolabelled primers and distinct dideoxynucleotides (ddNTPs), chemical analogues of deoxynucleotides (dNTPs). These ddNTPs acted as stoppers, preventing further extension of the DNA chain and resulting in fragments of varying lengths. Subsequently, these reactions were subjected to polyacrylamide gel electrophoresis to deduce the specific nucleotide sequence [11].

In a later advancement, the use of radioactive labels was replaced with fluorescent dyes to mark DNA fragments, and capillary electrophoresis emerged as a detection method. This innovation streamlined DNA sequencing and paved the way for automated sequencing machines [12]. Applied Biosystems (Waltham, MA, USA) introduced the ABI 370a in 1986, marking the debut of the first commercially available four-color fluorescence automated DNA sequencer. Following this milestone, in 1996, the ABI Prism 310 was introduced as the first capillary DNA sequencer [13].

Today, Sanger sequencing is still widely employed, particularly when the sequencing of individual genes or gene segments is required. It finds applications in viral or bacterial genotyping and the investigation of Single Nucleotide Polymorphisms (SNPs) associated with specific genomic regions (e.g., SNPs linked to antibiotic resistance) [14]. 

Sanger sequencing remains cost-effective for single-gene or targeted sequencing and is easily integrated into clinical laboratory workflows. Nevertheless, Sanger sequencing exhibits several limitations in comparison to modern NGS technologies [15]. It is relatively time-consuming and lacks the high-throughput capabilities of NGS, making it less suitable for the comprehensive sequencing of entire microbial genomes or extensive sets of genes. Additionally, it may not be optimal for detecting mixed infections or complex genetic variations.

### 2.2. NGS, or Second-Generation Sequencing

Unlike Sanger sequencing, next-generation sequencing (NGS) enables the simultaneous execution of billions of sequencing reactions, resulting in profound enhancements in throughput and cost-effectiveness (Table 2).

One of the pioneering methods used to replace the conventional Sanger approach was pyrosequencing, a sequencing-by-synthesis (SBS) technique reliant on the detection of pyrophosphate (PPi) release and the emission of light caused by nucleotide incorporation into a growing DNA chain [16]. Pyrosequencing technology generates an abundance of sequence reads in a single run, resulting in remarkably deep sampling. It presents potential advantages in terms of accuracy, flexibility, parallel processing, and automation feasibility [17]. Pyrosequencing has found widespread use in bacterial species identification, differentiation, and the detection of genetic mutations associated with antimicrobial resistance [18].

Ion semiconductor sequencing, another “sequencing by synthesis” method, operates on the principle of detecting hydrogen ions released during DNA polymerisation. What differentiates this technology from other SBS approaches is its unique characteristic of not relying on modified nucleotides or optical (fluorescence) detection methods [19]. Thermo Fisher Scientific’s (Waltham, MA, USA) ION Torrent is based on ion semiconductor sequencing, employing a semiconductor-based sequencing strategy. This approach involves the real-time detection of hydrogen ions (protons) released during DNA polymerisation, facilitating the rapid determination of DNA sequences [19]. Ion Torrent allows the swift and precise identification of diverse pathogens directly from clinical specimens [20], thereby reducing diagnostic turnaround times. Furthermore, it detects genetic variants with high precision, a critical factor in identifying infections and antibiotic resistance determinants. Ion Torrent is more cost-effective than other sequencing systems, rendering it suitable for various clinical settings. Nonetheless, limitations of this technology include shorter read lengths in comparison to other sequencing platforms, potential errors in sections with lengthy repetitions of the same base (homopolymers), and a higher error rate [21].

The bridge amplification method of sequencing, initially developed by Solexa (Hayward, CA, USA) and later acquired by Illumina (San Diego, CA, USA), has emerged as the most prominent and successful among the various NGS techniques. The Illumina sequencing platform is based on the ‘cyclic reversible termination’ method [22]: DNA libraries are created through random fragmentation of the DNA. These DNA fragments are then linked to adapters at both ends, followed by denaturation and immobilisation onto a solid surface known as a flow cell. On the flow cell, a series of steps, including bridge amplification and sequencing, take place to decode the genetic information. Illumina sequencing has gained paramount popularity and represents the most widely employed NGS technology to date [23]. It is renowned for its exceptional accuracy, producing high-quality sequencing data with error rates typically below 1%. This level of accuracy is crucial for applications such as variant detection. Illumina platforms support paired-end sequencing, delivering high-quality sequence data with comprehensive coverage and a substantial number of reads, supplying more information with reduced background noise compared to other second-generation techniques. However, one notable limitation of Illumina sequencing is the relatively short read lengths it produces, typically ranging from 150 to 300 base pairs. This can pose challenges for specific applications, such as de novo genome assembly and resolving repetitive regions within genomes, where longer read lengths are often necessary. Identifying structural variants, including large insertions, deletions, inversions, and translocations, can also be challenging with Illumina sequencing alone. Short reads, conversely, excel in targeted sequencing and profiling specific genomic regions or genes [24,25]. 

### 2.3. Third-Generation Sequencing

The third generation of sequencing technologies, also referred to as long-read sequencing (LRS) (Table 2), is based on the principles of single-molecule (SM) and real-time (RT) sequencing [26]. SM sequencing eliminates the need for DNA amplification, resulting in increased throughput, accelerated turnaround times, and extended read lengths. Nonetheless, LRS technologies do exhibit a relatively elevated raw error rate, approximately in the range of 10–13%, which is a noteworthy limitation. However, the employment of circular templates in LRS enables the sequencing of molecules spanning up to 1–2 kb multiple times, ultimately enhancing the overall accuracy [27].

In 2011, Pacific Biosciences (PacBio; Menlo Park, CA, USA) introduced the pioneering LRS method known as Single Molecule Real Time (SMRT) sequencing. SMRT sequencing enables the sequencing of exceptionally long fragments, extending up to 30–50 kb or even longer [28]. This approach involves fluorescence-labelled nucleotides, which are integrated by a polymerase positioned at the base of a densely packed array of zero-mode wavelength (ZMW) nanostructures. This configuration permits the real-time detection of fluorescence signals from millions of molecules simultaneously [29]. 

Long reads are invaluable for resolving complex genomic regions and detecting structural variants. Real-time data acquisition facilitates the monitoring of sequencing progress and applications such as swift pathogen detection. Notably, PacBio sequencing (as well as nanopore sequencing, discussed below) obviates the need for PCR amplification of the DNA, thereby mitigating biases introduced by PCR. This quality is particularly advantageous for GC-rich or highly repetitive genomic regions. Both PacBio and Nanopore technologies can operate with relatively small amounts of nucleic acid, rendering them suitable for studies involving limited sample material, including precious clinical specimens or low-abundance environmental samples. Moreover, PacBio’s technology boasts the capability to identify nucleotide modifications like methylation, making it an invaluable asset for epigenetic investigations. Furthermore, when combined with other techniques, such as Illumina sequencing, PacBio sequencing allows for the assembly of complete bacterial genomes with enhanced accuracy [30]. It is worth noting that PacBio sequencing exhibits higher raw error rates when compared to short-read sequencing platforms like Illumina. To attain a similar level of precision, error correction methods are often necessary, making it a challenge. Despite cost reductions, PacBio sequencing may still be relatively expensive, particularly for large-scale applications.

Oxford Nanopore Technologies (ONT), based in Oxford, UK, introduced its sequencing method in 2014 when it launched the MinION device. This technology generates long reads by employing nanopores embedded in a membrane, through which an ionic current flows [31]. In this approach, an electric field forces single-stranded molecules through a nanopore with a diameter of 2 nm, leading to the generation of distinct electric signals [32]. Since the length of nanopore reads is theoretically determined solely by the length of the DNA molecules being sequenced, if the DNA template is of sufficient quality, it should possible to obtain extremely long reads, covering hundreds of thousands of bases [27]. These long reads are particularly advantageous for resolving complex genomic regions, such as repetitive sequences and structural variants, and characterising mobile genetic elements, such as plasmids. Nanopore sequencing also enables direct RNA sequencing and provides real-time data acquisition, facilitating the monitoring of the sequencing process. These features are highly beneficial for rapid insights, including pathogen detection and identification. Similar to PacBio sequencing, nanopore sequencing can detect DNA and RNA modifications, such as methylations, which are crucial for epigenetic studies. To date, one of the primary challenges associated with ONT’s technology has been its relatively high basecalling error rate. Even with the recent introduction of basecallers based on deep learning algorithms, the error rate has decreased to a median value of approximately 5% [33]. Nanopore sequencing can be enhanced through the incorporation of complementary short-read data for error correction, particularly in regions with high GC content or repetitive sequences.

## 3. Genomic Analyses in Clinical Microbiology

### 3.1. Whole Genome Sequencing

Whole genome sequencing (WGS) is the process of determining and assembling the complete genetic code of an organism. It relies on NGS technologies, which can be categorised as second generation (e.g., Illumina) or third generation (e.g., PacBio or Nanopore). WGS is commonly employed for analysing the genomes of single bacterial isolates, but its application extends to difficult-to-culture microorganisms, such as *Mycobacterium tuberculosis*. 

Since the publication of the first bacterial genome of *Haemophilus influenzae* type B in 1995 [34], WGS has gained considerable prominence in the field of public health. This versatile technique serves multiple purposes, including organism identification, strain typing, and the detection of potential antimicrobial resistance genes, and contributes to epidemiological surveillance, such as by tracking vaccine-preventable diseases. It also plays a crucial role in assessing relatedness among strains during outbreak investigations, supporting hospital prevention programs, and monitoring the environment [35,36]. Although Sanger sequencing remains prevalent in clinical practice, WGS provides a more comprehensive analysis of a pathogen’s virulence and resistance profile.

WGS in microbiology has unquestionably ushered in a new era of high-resolution microbial genomics, allowing for in-depth analyses of genetic material within microbial populations. However, deriving meaningful insights from WGS data in this context presents its own set of bioinformatics challenges. These challenges encompass data quality control, accurate genome assembly, and effective gene annotation, particularly when dealing with the remarkable genetic diversity of microorganisms. Addressing these bioinformatics challenges demands robust and powerful analytical tools as well as well-trained personnel [37].

In microbiology, genome-wide association studies (GWAS) represent a relatively recent and exciting approach. The first successful application of GWAS to bacteria was published in 2013 [38,39]. Similar to human GWAS, these methods aim to establish statistical associations between genetic variations and observable traits within a population. In microbial studies, GWAS helps to establish links between the genotypes of microorganisms and their specific characteristics. Beyond genotyping [40], GWAS can assist in the identification of genetic factors associated with drug resistance or virulence [23,41,42]. Nonetheless, conducting GWAS on bacterial populations presents unique challenges that require careful consideration and innovative strategies [42,43]. Bacterial populations are genetically diverse and complex, due to factors like gene transfer and adaptation, making it difficult to identify associations between genetic variants and phenotypic traits. Thus, researchers must prioritise the selection of a diverse set of isolates and employ specialised analytical methods [44]. The presence of genetic variants with subtle effects on phenotypic traits poses another challenge, especially when dealing with limited sample sizes. This can be addressed by increasing the sample sizes and applying meta-analysis techniques to enhance the detection of associations [45]. Integrating GWAS results with functional genomics data, like transcriptomics or proteomics, can elucidate the biological mechanisms of observed associations [46]. Collaboration and data sharing among different research groups and institutions are essential for the success of bacterial GWAS studies. Open-access databases and platforms for sharing genetic and phenotypic data play a pivotal role in promoting collaboration and facilitating knowledge exchange [44]. 

In a recent study using a GWAS approach [47], genetic determinants linked to host specificity in *Escherichia coli* (*E. coli*) were thoroughly examined. The investigation revealed a range of distinct genetic factors that could facilitate *E. coli*’s adaptation to diverse host species, including the identification of a novel gene cluster, *nan-9*, which is notably conserved and prevalent in extraintestinal pathogenic *E. coli* (ExPEC) lineages associated with humans. The authors hypothesised that the presence of the *nan-9* gene cluster is a significant contributor to the adaptation of ExPEC to the human intestinal environment. In another study [48], researchers investigated the connection between genetic variations in *E. coli* and patient outcomes in bloodstream infections, through two potential entry points: the urinary or digestive tracts. Surprisingly, no significant associations were found between genetic variants and patient outcomes. This observation suggests that host-related factors play a predominant role in shaping the course of bloodstream infections. However, the researchers found a robust correlation between the *papGII* operon and the entry of *E. coli* through the urinary tract. This finding underlines the effectiveness of employing bacterial GWAS in real clinical scenarios, shedding light on the intricate interplay between bacteria and the human body.

### 3.2. Targeted Sequencing

Targeted sequencing (tNGS) is a highly sensitive and powerful method used for the precise detection and screening of genetic variants and mutations, within specific genomic regions. It has evolved into an indispensable and routine technique in both clinical and research domains, particularly in the realms of cancer research and the investigation of various human diseases [49]. In tNGS, a selection or enrichment process is applied either before or after library preparation, with a focus on specific groups of microorganisms or even individual microorganisms. This selection is typically accomplished through multiplex-PCR amplification or probe hybridisation methods. In contrast to WGS, which analyses the entire genome of an organism, tNGS is customised to examine specific genomic regions of interest. This targeted approach offers several advantages and presents unique challenges [50]. One of the key advantages of tNGS is its exceptional sensitivity in detecting variants and mutations within predefined genomic regions. This heightened sensitivity holds particular significance in clinical applications, where even low-frequency mutations can exert a substantial impact on disease diagnosis, prognosis, and treatment selection. Notably, tNGS finds extensive applications in clinical virology diagnostics. For instance, in HIV research, tNGS can identify minor variants carrying drug resistance mutations that might elude detection with traditional Sanger sequencing [24]. Similarly, tNGS has played a pivotal role in characterising variants of the SARS-CoV-2 virus during the COVID-19 pandemic, aiding in the tracking of specific strain spread and the assessment of the potential implications for diagnostics, vaccines, and therapeutics [51,52]. In microbial identification, tNGS is particularly valuable for distinguishing bacterial species using the 16S ribosomal RNA (rRNA) gene or fungal species through the internal transcribed spacer 1/2 (ITS-1/2) regions. This approach enables precise taxonomic classification and can be applied to various environmental samples, clinical specimens, and food safety assessments [53,54,55]. However, one substantial challenge in tNGS is the meticulous selection of target regions. The design of primers or probes for enrichment necessitates an in-depth understanding of the genomic characteristics of the organisms under investigation. An inaccurate selection of target regions can result in incomplete or biased results [56,57]. Furthermore, data analysis and interpretation can be complex, especially when dealing with diverse microbial populations. The differentiation between closely related species or strains may require specialised bioinformatics tools and reference databases [58]. Therefore, it is essential to implement proper quality control measures to ensure the reliability of tNGS results. 

A programmatic model for the implementation of tNGS has been put into action in Namibia, an upper middle-income country located in Southern Africa. Namibia faces a significant burden of tuberculosis (TB) [56]. The overarching goal of integrating tNGS into routine practices in high-TB-incidence countries, such as Namibia, is to enhance the clinical management of TB cases and establish effective surveillance mechanisms for resistance to new treatment regimens. The cost-effectiveness and enhanced efficiency associated with tNGS, compared to other sequencing techniques, make this approach theoretically ideal for regions with limited resources. In 2018, scientists from the Plasmodium Diversity Network Africa (PDNA) explored the potential of using tNGS for genetic studies and malaria monitoring across Africa [59]. They also identified how tNGS could expedite malaria research, advance science, and improve public health in sub-Saharan Africa. However, they recognised several challenges that must be overcome, such as securing research funding, establishing the necessary infrastructure, and developing a skilled workforce.

### 3.3. Metagenomics

While WGS examines the complete genetic makeup of a single bacterial colony, metagenomics, whether amplicon-based or next-generation sequencing metagenomics (mNGS), is designed to unravel the complexities of microbial communities within a given sample, often without the prerequisite of prior cultivation.

Amplicon-based metagenomics typically relies on PCR to selectively amplify specific genetic markers. Among these markers, the 16S rRNA gene stands out as a widely embraced target due to its high conservation across bacterial species [60]. The integration of NGS technologies into amplicon sequencing of the 16S rRNA gene has elevated the precision and sensitivity of microbiome studies. This approach not only permits the detection of less abundant and challenging-to-culture microorganisms, but also contributes to a reduction in the overall analysis costs [61]. 

The role of microbiota in health and disease has garnered increasing attention following its discovery [62]. Dysbiosis, characterised by imbalances in the microbiota, has been linked to conditions like inflammatory bowel disease, allergies, autoimmune disorders, and even cancer development and progression, particularly in the gut. Individuals with inflammatory bowel disease and colorectal cancer often exhibit decreased bacterial diversity and abundance compared to healthy individuals, with a prevalence of *Firmicutes* and *Bacteroidetes*. Specific bacteria such as *Fusobacterium nucleatum*, *E. coli*, *Enterococcus faecalis*, *Streptococcus gallolyticus*, and *Bacteroides fragilis* have been associated with the onset and progression of colorectal cancer [63].

Emerging studies have unveiled the gut-brain axis, revealing potential links between gut microbiota and conditions like depression, anxiety, and neurodegenerative diseases. An interesting study [64] combined 16S rRNA sequencing with structural magnetic resonance imaging and resting-state functional MRI to explore differences in faecal microbiota between patients with schizophrenia (SZ) and controls (NC). The findings showed a significantly lower relative abundance of *Ruminococcus* and *Roseburia* in SZ patients, and a significantly higher abundance of *Veillonella*, compared to NCs. Furthermore, the study revealed significant correlations between the gut microbiome and brain structure and function in SZ patients, suggesting that the gut microbiome characteristics could be linked to alterations in brain structure and function.

However, it is important to acknowledge a limitation of amplicon-based sequencing [50,65]: it primarily provides taxonomic information, identifying the presence and abundance of various bacterial species within a sample. While this is invaluable for characterising microbial communities, it does not directly reveal the functional attributes or metabolic potential associated with these species. Attempting to infer functional insights solely from genomic sequences of reference strains may lead to erroneous conclusions. This is because genomes and their functional attributes can vary significantly among closely related species and strains in different environmental or contextual settings. 

An evolution of amplicon-based metagenomics is represented by shotgun metagenomics, often referred to as metagenomic next-generation sequencing (mNGS). This is a potent method for the in-depth exploration of microbial communities, enabling the comprehensive sequencing of entire genomes within these communities without the constraints of prior cultivation. This breakthrough has revolutionised the ability to investigate previously unculturable or unknown microorganisms. In pathogen detection, mNGS have demonstrated remarkable advantages, encompassing a broad spectrum of infectious agents, from bacteria and mycobacteria to viruses, yeasts, and parasites [66]. Beyond this pivotal role, mNGS has extended its scope to comprehensively explore microbial ecosystems, including those residing within the human body or the environment [24,67]. Notably, it has expanded its application to specimens once considered sterile, such as joint fluid, cerebrospinal fluid, or blood [68,69,70,71]. Furthermore, in clinical settings, mNGS is emerging as a promising tool for diagnosing infectious diseases that are either rare or extraordinarily complex [72]. 

In a recent cross-sectional study [73], mNGS was employed for the diagnosis of pulmonary infection by *Tropheryma whipplei*. This bacterium is primarily associated with Whipple’s disease, a chronic systemic infectious disease primarily involving the gastrointestinal tract but posing a potential risk for pneumonia when detected in immunocompromised patients. The use of mNGS considerably increased the number of positive cases detected in bronchoalveolar lavage fluid. mNGS has also found valuable applications in the direct detection of antimicrobial resistance genes within clinical samples. An illustrative example of this application is the detection of carbapenemase-producing *Enterobacterales* (CPE) from rectal swabs, a critical aspect of infection control efforts [74].

Even though initial efforts have focused on pathogen detection, mNGS may be useful in detecting antibiotic resistance and virulence factor genes directly from clinical samples, provided that adequate coverage is available [66]. Traditionally, identifying the sources of antimicrobial resistance (AMR) relied on isolating pathogens. However, metagenomics has opened new avenues for studying AMR determinants comprehensively. In a recent study [75], researchers employed shotgun metagenomics to analyse faecal samples from livestock across Europe and from humans with occupational exposure to these animals. Their findings revealed both country-specific and universal AMR determinants. Moreover, the study assessed how the presence of country-specific determinants affected the attribution of AMR resistance in humans.

Despite its promise, mNGS presents several challenges. One major limitation is the potential for background noise, primarily stemming from human nucleic acids or the resident microbiome. This issue can be particularly pronounced in specimens with low pathogen loads or when analysing complex microbial communities [76]. This challenge can be especially concerning in specimens such as tissues or respiratory secretions, or when the pathogen load is exceptionally low [77]. Addressing this requires robust bioinformatics pipelines capable of accurately distinguishing relevant microbial sequences from noise. However, the computational analysis of mNGS data presents a formidable challenge. While numerous open-source and commercial software packages are available for mNGS data analysis [78], selecting the appropriate algorithms and parameter settings can significantly impact the accuracy and interpretability of results. Researchers often grapple with the complexity of these analyses, underscoring the need for expertise in bioinformatics.

In summary, shotgun metagenomics, or mNGS, offers a powerful approach to studying microbial communities, but challenges related to bioinformatics analysis must be carefully addressed.

An overview of the diverse approaches to genomic sequencing is presented in Table 3, delineating the distinct methodologies and common applications associated with each technique.

## 4. Use of Genomic Approaches to Detect Antimicrobial Resistance

### 4.1. Preface

Antimicrobial resistance is one of the top ten global public health threats, as identified by the World Health Organization (WHO) [79]. 

A recent study [80] examined the global mortality estimates for thirty-three different bacteria in 2019. This study found that five pathogens—*Staphylococcus aureus*, *Escherichia coli*, *Streptococcus pneumoniae*, *Klebsiella pneumoniae*, and *Pseudomonas aeruginosa*—accounted for approximately 55% of deaths. The same study, when evaluating the global burden of AMR, estimated 4.95 million bacterial AMR-related fatalities in 2019. This included 1.27 million bacterial AMR-related deaths, with the six primary pathogens responsible for deaths due to resistance (*E. coli*, *S. aureus*, *K. pneumoniae*, *S. pneumoniae*, *Acinetobacter baumannii*, and *P. aeruginosa*), and identified as priority pathogens by the WHO, contributing to 929 thousand deaths attributed to AMR and 3.57 million deaths associated with AMR [81].

Additionally, in 2020, compared to 2017, bloodstream infections caused by resistant *E. coli*, *Salmonella* species, and *Neisseria gonorrhoeae* strains increased by at least 15%. Resistance levels in *K. pneumoniae* and *Acinetobacter* species exceeded 50%, and 8% of *K. pneumoniae* infections were resistant to last-resort antibiotics like carbapenems [82]. The European Centre for Disease Prevention and Control (ECDC), in its latest report on antimicrobial resistance surveillance [83], noted that 33% of European countries reported resistance percentages in *K. pneumoniae* at 25% or higher. Furthermore, carbapenem resistance was prevalent in both *P. aeruginosa* and *Acinetobacter* species, surpassing even higher percentages than in *K. pneumoniae*.

### 4.2. Status and Perspectives

Clinical laboratories still rely on culture-based antimicrobial susceptibility testing (AST), although phenotypic methods (e.g., broth microdilution and disk diffusion) are often complemented with rapid molecular methods (e.g., PCR for resistance determinants). Notably, despite the potential of NGS to rapidly, consistently, and accurately predict resistance in microbial strains or populations, by examining the entire resistome and providing surveillance data, the utilisation of NGS for susceptibility prediction remains relatively uncommon. Only a limited number of clinical microbiology labs possess the financial funds and trained staff to access this advanced technology.

The adoption of genome-based resistance detection in clinical microbiology can yield substantial benefits. Presently, culture-based methods require approximately 24 to 48 h for microbial identification using matrix-assisted laser desorption-ionisation-time-of-flight mass-spectrometry (MALDI-TOF) and an additional 48 to 72 h for reporting AST results [84]. In contrast, mNGS, being culture-independent, has the potential to reduce this turnaround time to a remarkable 6–8 h, irrespective of the growth rates. This can be particularly advantageous when dealing with slow-growing pathogens or fastidious bacteria [85,86]. For example, in the case of *M. tuberculosis*, WGS -based diagnostics significantly reduced the time to the confirmation of TB diagnoses and provided accurate drug resistance profiles [7]. However, integrating WGS data into clinical practice poses complex challenges [87]. The reliable identification of AMR genes depends on robust bioinformatics tools and expertise [88,89]. Understanding the genetic variants and their implications for resistance phenotypes can be demanding, particularly for those lacking specialised knowledge. WGS can unveil novel resistance mechanisms not covered by existing databases, but identifying and characterising these new mechanisms can be time-consuming and may require functional validation. To guide effective treatment decisions, it is imperative to integrate WGS results with clinical data, including patient history, antibiotic susceptibility testing, and treatment outcomes. Achieving this integration can be challenging, necessitating interdisciplinary collaboration and the adoption of standardised data formats [90]. Interlaboratory variability can affect the accuracy of AMR gene detection, making the standardisation of WGS protocols and data analysis pipelines across laboratories vital to ensure consistent and reproducible results [85]. While the cost of WGS has decreased over time, it may still pose a barrier for some clinical laboratories. The utilisation of WGS data in clinical decision-making raises pertinent regulatory and ethical concerns, such as those regarding data privacy and consent. Establishing clear guidelines and ethical frameworks is essential in this context [91].

Both amplicon and shotgun metagenomics can also detect AMR factors. Compared to conventional methods, these metagenomic techniques offer faster results and higher throughput. Notably, mNGS, which does not rely on amplification, has the potential to identify all pathogens in a sample and simultaneously detect and quantify thousands of AMR genes without prior selection, addressing all elements involved in resistance acquisition [92]. Improved computational methods, along with novel bioinformatic tools and AMR-determinant databases, may serve as complementary tools for culture-based methods. WGS is particularly useful for elucidating the elements involved in the evolution of AMR and for understanding novel resistance mechanisms. Furthermore, genomic data can be easily accessed for other purposes, such as phylogenetic and surveillance studies [93]. Real-time AMR surveillance through WGS could aid in the early detection of outbreaks and the identification of their sources, thus supporting public health decisions and policies.

Despite sequencing technologies becoming more cost-effective, the time and resources required for genome analysis remain significant barriers to their use in clinical routine settings [94]. While Illumina is currently the dominant platform, optimising turnaround times and batching multiple samples for efficiency is crucial to improve cost-effectiveness. Long-read technologies, such as Nanopore, may enable the faster and less expensive sequencing of fewer isolates [95].

A European Committee on Antimicrobial Susceptibility Testing (EUCAST) Subcommittee addressed the relationship between phenotypic antimicrobial susceptibility testing and WGS [96]. One significant limitation of using NGS technologies in AMR analysis is that only known resistance genes and mutations, or very similar genetic variants, can be detected. As a result, the genomic prediction of AMR phenotypes can be accurate only for well-characterised bacterial species and AMR determinants [97]. The key challenge, however, is in identifying the chromosomal alterations that cause changes in gene expression. Genomic methods can detect the presence of resistance determinants but not their expression, so they cannot provide the minimal inhibitory concentration (MIC) of a compound. They may also overestimate resistance and provide discordant results when compared with phenotypic susceptibility results [96,98].

Recent advancements in bacterial gene expression analysis involve cutting-edge technologies such as RNA Sequencing (RNA-Seq), Single-Cell RNA Sequencing (scRNA-Seq), and Mass Spectrometry-Based Proteomics for direct protein measurement. These methods provide valuable insights into the dynamic mechanisms behind AMR [46]. Computational methods enable the integration of this multiomics data, combining genomics, transcriptomics, and proteomics to offer a comprehensive view of AMR mechanisms [93]. These approaches have the potential to advance our understanding of AMR and improve the accuracy of resistance predictions in clinical and research settings. However, several challenges must be considered, including managing complex data integration, dealing with limited information for specific bacteria and resistance types, and accounting for biological variations. Therefore, rigorous validation and interpretation are essential when working with multiomics predictions [46,99,100].

Table 4 provides a summary of the advantages and disadvantages associated with genomic methods used for the detection of AMR.

It is universally recognised that AMR is a critical and prevalent worldwide threat. To address it effectively, a coordinated and standardised approach to epidemiological surveillance should be employed. Today, AMR surveillance is primarily conducted through traditional phenotypic techniques. However, as outlined in Section VI, titled “*Genomic surveillance of infectious diseases and dissemination of antimicrobial resistance*”, there is a growing need to combine genomic surveillance approaches, notably WGS, with phenotypic and epidemiological data. The integration of genomic surveillance into the existing framework is driven by the recognition that it can significantly enhance the ability to monitor and combat AMR effectively.

## 5. Use of Genome Sequencing in Hospital Outbreak Investigations

Hospital outbreaks can have severe implications for patient well-being, resulting in extended hospital stays and increased healthcare costs. Traditionally, investigating such outbreaks has involved complex, pathogen-specific procedures, often limited to a few reference and public health laboratories. These methods, however, lack precision and can be time-consuming [101]. In contrast, genomic characterisations provide diagnostic microbiology laboratories with the ability to trace shared exposures back to a single source of infection. Promptly identifying the source of an outbreak can aid in its containment. Genomic investigations of outbreaks are primarily retrospective, conducted after an outbreak is suspected or resolved. Nevertheless, due to their exceptional resolution, genotyping methods employing WGS represent the most effective means of unveiling genetic relationships and informing infection prevention and control (IPC) strategies [102]. IPC teams can also employ WGS to rule out outbreaks, thus averting service disruptions and enabling healthcare personnel to concentrate on preventive measures [103]. A relatively straightforward method for tracking bacterial transmissions and outbreaks, known as “reverse genomic epidemiology”, has recently been proposed [102,104]. According to this approach, genomic data obtained from WGS are analysed to compare the genetic profiles of the isolates. If multiple isolates exhibit highly similar genetic profiles and cluster together on the phylogenetic tree, it suggests that they likely originated from a common source of infection, such as contaminated medical equipment or surfaces. This strategy has also been applied to investigate community outbreaks of infectious diseases. By analysing the genetic relatedness of isolates from multiple patients, public health officials can identify the source of the outbreak, such as a contaminated water supply or a shared public venue. Epidemiologists can then directly identify similar isolates in online databases (e.g., the PulseNet database for foodborne diseases [105]), reducing reliance on other epidemiological evidence [104]. In a recent study conducted in Sweden [106], the genetic relatedness of *Staphylococcus haemolyticus* isolates revealed the presence of a clonal outbreak strain that had emerged in the 1990s within a neonatal intensive care unit. Notably, an identical outbreak strain had also been identified in separate instances in Japan and Norway, indicating its dissemination across geographic regions and over time. Similarly, through genome-wide analysis, it was observed that NDM-1-producing *K. pneumoniae* ST11 strains with identical genetic characteristics were consistently spreading across various wards within a hospital in Portugal [107]. These data guide tailored treatment strategies and infection control measures.

Once isolates from an outbreak have undergone WGS, two primary approaches can be employed to establish relationships between them: constructing phylogenies based on Single Nucleotide Polymorphisms (SNPs) or extended gene-by-gene comparison with core genome Multilocus Sequence Typing (cgMLST) [108]. The first method involves building phylogenies by assessing relatedness through SNP distances between isolates. SNPs are evaluated by comparing sequence reads to a closely related reference genome and identifying nucleotide differences. Only positions in the core genome, covered by all query genomes, are used to establish a set of core SNPs [109]. An SNP distance matrix is then computed for all pairwise combinations, enabling subsequent phylogenetic analysis [110]. In contrast to conventional Multilocus Sequence Typing (MLST), which examines genetic similarity based on only seven genes, core genome MLST assesses thousands of gene regions or alleles. This is achieved by aligning complete or draft genome assemblies with a species-specific database of allelic variants [111]. Because no outbreak-specific reference is required, cgMLST is a suitable and unbiased method for identifying potential clusters. One major disadvantage of cgMLST is that results obtained by different laboratories may not be directly comparable. However, this issue can be mitigated by the real-time synchronisation of the local allele database with a centrally curated cgMLST allele nomenclature server [112,113].

Another innovative approach relies on k-mer typing [114]. K-mers are contiguous substrings of length k in a given string (any string sequence: DNA, RNA, protein). K-mers are contiguous substrings of length k in a given string (any string sequence: DNA, RNA, protein). The k-mers act as molecular markers, capturing valuable information about a microorganism’s genetic makeup. Scientists may quickly and accurately identify the species or genus of an unknown bacterium by comparing its k-mer profiles to a reference database of k-mer profiles from known species, even if the organism has never been cultivated or characterised. While k-mer approaches are predominantly used for microbial taxonomy studies, they can also be applied in sub-clustering and identifying isolates with known genomes [115]. K-mers allow for the assessment of the relatedness and differentiation between different strains or isolates within a species. In outbreak studies, by comparing the k-mer profiles of isolates from different patients or sources, it is possible to determine whether they share a common source of infection. This information is also crucial for understanding the evolution and epidemiology of microbial pathogens.

In a recent study [116], ore SNP-analysis and reference-free split k-mer analysis (SKA) were employed to track the transmission of Vancomycin-Resistant *Enterococcus faecium* (VREfm) in routine outbreak data. Both methods exhibited superior discriminatory power and displayed robust associations with suspected local hospital outbreaks and systematic epidemiological categorisations. Notably, K-mer analysis, specifically SKA, demonstrated the highest degree of correlation with both outbreak data and epidemiological information.

## 6. Genomic Surveillance of Infectious Diseases and Dissemination of Antimicrobial Resistance

Genomic surveillance has emerged as the leading methodology for the analysis, monitoring, and control of pathogens. This cutting-edge approach facilitates the rapid and precise examination of microbial genetic material, enabling the swift detection of outbreaks and emerging threats, and a comprehensive understanding of pathogen epidemiology, including modes of infection transmission. Additionally, genomic surveillance data can significantly strengthen national AMR surveillance systems. The value of genomic surveillance for public health has been vividly demonstrated during the COVID-19 pandemic [117]. It is worth noting that at the outset of the SARS-CoV-2 pandemic, no national healthcare system possessed adequate control measures or surveillance mechanisms to respond swiftly to emerging infectious diseases [118].

Despite considerable progress in sequencing technologies and data analytics, which allow for high-throughput and cost-effective pathogen detection and typing [119,120], their full potential in infectious disease surveillance has yet to be fully realised [121,122,123]. The MiSeq platform is widely used for infectious disease surveillance and research. It offers consistent and reliable performance, producing high-quality sequencing data, adaptability, user-friendliness, and cost-effective solutions without compromising data quality. However, the continuous evolution of sequencing technologies, including the development of novel flow cells, presents an intriguing prospect for integrating the MinION platform into conventional surveillance methodologies [124]. The MinION platform, known for its portability and scalability, holds promise for addressing specific challenges, especially in smaller laboratories with limited resources. Its ability to deliver rapid and cost-effective sequencing has the potential to enhance the diagnostic capabilities of these facilities, contributing to more effective surveillance practices and expedited responses to infectious disease threats. Establishing and maintaining genomic surveillance capabilities in resource-limited settings is a complex effort filled with challenges [125,126]. It demands a considerable pool of expertise, encompassing both wet laboratory techniques (such as DNA extraction and sequencing) and dry laboratory skills (including data analysis and interpretation). Ensuring that personnel possess these specialised skills can be particularly challenging in settings where resources for training and education are limited. Moreover, the integration of specialised information technology (IT) infrastructure is crucial. Genomic surveillance generates massive volumes of data that require efficient storage, processing, and analysis. Implementing and maintaining the necessary IT systems can be resource-intensive and may require technical support that is often scarce in resource-limited environments. Additionally, establishing quality management systems that adhere to the rigorous standards of public health laboratories is essential.

In 2022, the World Health Organization (WHO) initiated a 10-year strategy aimed at broadening the application of genomics for monitoring and responding to public health crises. The objective is to establish genomic surveillance as a key tool in preparedness and rapid response to pandemics [127]. Similarly, in May 2023, the WHO introduced the Pathogen Surveillance Network (IPSN), a global network of organisations with expertise in pathogen genomics, designed to “*accelerate progress in the deployment of pathogen genomics and improve public health decision-making*” [128].

Genomic sequence data from representative populations are valuable for monitoring viruses, enabling the detection of new variants and tracking trends in existing ones. In recent years, WGS has been increasingly used for viral genomic epidemiology [129]. The significance of genomic data became evident during the SARS-CoV-2 pandemic. Genomic information played a crucial role in national surveillance systems, effectively controlling the spread of new viruses and identifying both established and novel variants. As a result, many countries that had not previously utilised genomic data began to integrate them into their surveillance efforts [130]. Also, viral genomic data are of paramount importance in the development of vaccines and antiviral treatments. They are instrumental in ensuring that these treatments remain effective, especially when escape or resistance mutations emerge [3,131,132].

Influenza surveillance is crucial due to the limitations of seasonal influenza vaccines that provide only partial protection [133]. The constant threat lies in the potential emergence of a pandemic influenza virus, caused by antigenic shifts, as was the case with the H1N1 pandemic strain in 2009 [134]. Currently, the WHO and the European Centre for Disease Prevention and Control (ECDC) are primarily focusing their efforts on the genetic surveillance of the hemagglutinin (HA) gene, often the sole segment sequenced using the Sanger method [135]. With the increasing accessibility of NGS, obtaining sequences from all eight influenza segments concurrently has become a more cost-effective approach. The Centers for Disease Control and Prevention (CDC), which has been employing NGS methodologies since 2014, typically sequences the complete genomes of approximately 7000 influenza viruses annually through virologic surveillance. This extensive sequencing significantly contributes to informed public health decision-making [136]. In Europe, influenza surveillance systems differ from country to country [137], leading to substantial disparities in data quality practices among EU Member States [138]. The European Influenza Surveillance Network (EISN) supervises epidemiological and virological surveillance data. In the United Kingdom, the Respiratory Virus and Microbiome Initiative (RVI) was launched at the outset of 2023 [139]. Its primary objective is to establish the capacity for routine genomic surveillance of respiratory viruses, encompassing influenza, respiratory syncytial virus, adenovirus, and rhinovirus.

Antimicrobial resistance undeniably stands as a major global threat [81]. The development of novel drugs must go hand in hand with well-coordinated and standardised epidemiological surveillance efforts. Currently, AMR surveillance primarily relies on phenotypic characterisation. Nevertheless, genomic surveillance through WGS, while not yet capable of entirely replacing AST, is essential for the integration of phenotypic and epidemiological data [140]. In 2022, the Organization for Economic Cooperation and Development (OECD) identified pressing priorities for the European Union/European Economic Area (EU/EAA). These priorities include strengthening AMR surveillance by enhancing laboratory network capacity and incorporating new data sources and technologies, such as genomic surveillance through WGS [141]. Within the European Region, two regional networks gather and present AMR surveillance data from nearly all fifty-three member states: the European Antimicrobial Resistance Surveillance Network (EARS-Net) and the Central Asian and European Surveillance of Antimicrobial Resistance (CAESAR). The most recent EARS-Net reporting protocol, released in March 2023 [142], outlines a comprehensive list of bacteria and antimicrobial agent combinations, which will be monitored through epidemiological surveillance. What is particularly noteworthy is that, even though the majority of European countries have established national action plans to address AMR, a significant proportion, amounting to 16% of these countries, reported that they continue to collect AMR data at the local level without adhering to a standardised approach. This indicates that, despite the recognition of the AMR threat and the development of national strategies, there is still a need for more standardised and coordinated efforts to collect and analyse data related to antimicrobial resistance.

## 7. One Health Genomics and Perspectives

According to the latest definition, “One Health is an integrated, unifying approach that aims to sustainably balance and optimize the health of people, animals and ecosystems. It recognizes that the health of humans, domestic and wild animals, plants, and the wider environment (including ecosystems) are closely linked and inter-dependent” [143]. The One Health initiative aspires to establish comprehensive strategies for disease surveillance, control, and prevention. Its primary objective revolves around creating a unified system encompassing surveillance in humans, animals, and the environment [144]. This initiative involves investments in surveillance infrastructure for both human and animal health, promoting the timely exchange of information and fostering interdisciplinary collaborations [145,146]. Genomics plays a pivotal role in achieving these objectives. When integrated with epidemiological data from One Health surveillance systems, genomic data assist in predicting population expansions and disease transmission across species, enabling proactive measures before human health is jeopardised [145]. 

One Health genomics is crucial for controlling zoonotic diseases and AMR. The interconnectedness of humans, animals, and the environment has been disrupted due to factors such as intensive livestock farming, urbanisation encroaching on wildlife habitats, and the global trade in exotic animals. This disruption has heightened the risk of spillover events and the rapid emergence of new zoonotic diseases [147]. The COVID-19 pandemic has underscored the importance of preventing future pandemics resulting from emerging and re-emerging zoonotic diseases like monkeypox and dengue. This has led to widespread concern and a strong interest in preventive measures. Addressing the transmission of zoonotic diseases necessitates a multidisciplinary approach that encompasses disease genomics, epidemiological surveillance, and predictive modelling [148]. Effective surveillance necessitates a collaboration between the human health, animal health, and agricultural sectors. Effective surveillance demands collaboration between the human health, animal health, and agricultural sectors. This collaborative approach includes the coordinated testing of samples from humans, animals (wildlife, livestock, and domestic), the environment (soil and water), and food sources [149]. Such collaborations offer significant advantages [150]. By pooling insights and data, the ability to detect potential disease outbreaks or unusual patterns is significantly enhanced. Timely detection is crucial for implementing proactive interventions and minimising the impact of emerging threats. Resource optimisation is another key advantage. Collaborative efforts ensure the efficient allocation of resources for research, surveillance, or intervention measures. Collaboration is particularly essential when addressing complex issues such as antimicrobial resistance and food safety [151,152]. These challenges demand comprehensive solutions covering all aspects from production to consumption. Collaborative efforts enable the development and implementation of holistic strategies that can be more effective in protecting public health.

Genomics plays a pivotal role in One Health initiatives. Genomics offers a standardised approach to characterising pathogens, both known and unknown, across species and ecosystems. This standardisation is critical in a One Health context, as it ensures that data from diverse sources and sectors are compatible and can be readily shared and analysed. Genomics significantly contributes to early detection. By rapidly sequencing and analysing pathogen genomes, it enables the swift identification of potential threats [145,153]. Whether it is a novel zoonotic disease or a foodborne pathogen, genomics can pinpoint these issues at an early stage, enabling a faster response. Once a threat is identified, genomics aids in the development of targeted interventions, such as vaccines or treatments [154]. This is particularly valuable in cases where a pathogen can impact both humans and animals, ensuring that the response is appropriate for all affected parties.

Significant examples of pathogens with the capacity to affect both human and animal hosts include the influenza A viruses and *Salmonella* [155,156,157,158]. Influenza A viruses can infect a wide range of hosts, including humans, birds, and mammals. The genomic surveillance of these viruses helps in the early identification of genetic variants, which may pose heightened risks to human and animal health. The precise identification of genomic signatures not only allows for the timely initiation of preventive measures, but also for the development of vaccines that are effective in both human and animal populations. *Salmonella* can cause foodborne diseases in both humans and animals. Genomic analysis can reveal the genetic mechanisms of antimicrobial resistance within *Salmonella* strains. Furthermore, genomic insights can inform the design and implementation of stringent food safety measures, aimed at mitigating the transmission of antimicrobial-resistant strains.

In recent years, several national and international genomic surveillance systems following the One Health approach have been implemented. The PREDICT initiative [159], a groundbreaking One Health project, exemplifies the transformative potential of genomics in global disease surveillance. Launched by the United States Agency for International Development (USAID) in 2009, PREDICT aimed to detect and prevent emerging infectious diseases at the wildlife–human interface. Employing innovative genomics and viral discovery techniques, the project sought to monitor and characterise viruses circulating in wildlife populations that could potentially spill over into humans. This approach enables early warning and responses to pandemics.

The Global Initiative on Sharing All Influenza Data (GISAID) [160] represents a global effort that utilises genomics to enhance pandemic preparedness. Established in 2008 as a response to the H1N1 avian influenza outbreak, GISAID emphasises the importance of open and timely sharing of influenza genomic data. It has played a pivotal role in tracking the evolution and transmission of influenza viruses, including seasonal and pandemic strains and, most recently, SARS-CoV-2.

In Italy, the National Institute of Health (ISS) introduced the IRIDA-ARIES system for foodborne disease surveillance. This system integrates genomic data and metadata from both human and non-human isolates [161]. By harmonising information from multiple sectors, including clinical and environmental data, the IRIDA-ARIES system [161] seeks to reinforce surveillance, prevention, and risk management efforts to proactively prevent the occurrence of foodborne diseases. The system integrates genomic data and metadata from both human and non-human isolates. Similarly, Switzerland has launched the Swiss Pathogen Surveillance Platform (SPSP) [162,163]. This innovative platform serves as a centralised hub for the monitoring of microorganisms, extending its scope across humans, animals, and the environment. By employing genomics as a central tool, the SPSP seeks to provide valuable data that can inform early disease detection, track the spread of pathogens, and support the development of effective prevention and control measures.

Metagenomics [164,165], not relying on prior knowledge of genomic sequences, significantly improves surveillance systems’ capacity to detect rare zoonotic transmissions with pandemic potential. It is particularly valuable in early warning systems, facilitating the prompt identification of emerging infectious agents. Furthermore, metagenomics provides the capacity to track the genetic diversity and evolution of pathogens over time, aiding in the assessment of potential risks and the design of targeted interventions. However, the sizable volume of data generated necessitates advanced computational and bioinformatic capabilities for data processing, which can be resource-intensive. Additionally, the standardisation and harmonisation of methods across surveillance systems remain a challenge, given the evolving nature of metagenomics technologies.

In the context of surveillance initiatives based on metagenomics, The Global Virome Project was a large-scale effort to create a global atlas of potential zoonotic viral pathogens, increasing the ability to identify and detect viruses that could pose a threat to human health [166]. The Remote Emerging Disease Intelligence—Network (REDI-NET) [167,168] is an initiative aligned with the One Health approach, emphasising a comprehensive surveillance strategy focused on the metagenomic surveillance of high-risk diseases. This innovative approach is being implemented in the United States, Kenya, and Belize, with the primary goal of enhancing the ability to detect and respond to emerging diseases in remote and ecologically diverse regions. What distinguishes REDI-NET is its use of metagenomic surveillance, which can identify both known and previously unknown microorganisms, making it valuable when dealing with diseases with zoonotic potential, where pathogens can jump between animals and humans, and can occur in various ecological niches. REDI-NET’s surveillance is not limited to a single sample type; it encompasses a wide range, including water, ticks, soil, and leeches. This holistic sampling approach recognises that diseases can emerge from various sources, and thorough surveillance across diverse ecosystems is essential to detect potential threats early on.

In the One Health approach, where the interconnectedness of human health, animal health, and the environment is paramount, genomics plays a pivotal role in bolstering the response to the escalating crisis of AMR. Its practical application extends beyond conventional methodologies, providing invaluable insights into the dynamics of AMR across these domains [169,170]. When outbreaks or sporadic cases of resistant pathogens occur, genomics equips us with the ability to scrutinise the genetic fingerprints of these microorganisms. This effectively creates a genetic map of transmission, unveiling how resistance traverses between species, ecosystems, and geographical locations. On a global scale, the One Health model incorporates molecular epidemiological elements that significantly enhance our comprehension of the evolutionary dynamics and genetic relatedness of AMR in a large spectrum of entities, including pathogens, vectors, hosts (both human and animal), and the ambient environment [171]. To gain a profound understanding of the foundational aspects of AMR, it is imperative to advance our insights into the intricate interplay among humans, animals, and their ecological surroundings, a feat achieved through the high-resolution tools offered by genomics. Metagenomics, a critical component of this endeavour, serves as a key instrument in unveiling the intricate patterns governing the dissemination of antibiotic resistance genes across diverse ecological niches. Simultaneously, WGS of isolates sampled from a diverse array of reservoirs offers indispensable disclosures into the potential trajectories of transmission [172].

Multi-drug resistant (MDR) *E. coli* is a predominant source of infections in both hospital and community settings. To address the increasing prevalence of MDR *E. coli* infections in humans, it is essential to gain insights into the reservoirs and origins from which humans acquire these infections. A combination of WGS, epidemiology, and ecology was employed to investigate the prevalence of AMR carriage and to characterise the diversity of AMR genes in *E. coli*, in Nairobi, Kenya [173]. The study pointed out that the composition of AMR gene communities was not linked to the host species. However, AMR genes were frequently found to be co-located, which potentially facilitated the acquisition and dissemination of multiple resistance in a single step. In contrast, a study in the United Kingdom had very different results [174]. It found a low occurrence of shared antimicrobial resistance genes between livestock and humans. This conclusion was drawn from an analysis of mobile genetic elements and long-read sequencing. Core genome comparisons through phylogenetic analysis revealed clear genetic distinctions between isolates from livestock and those from patients. This suggested that *E. coli* strains responsible for severe human infections did not directly originate from livestock sources.

So, while numerous studies have highlighted similarities across different reservoirs, and have shed light on potential AMR transmission pathways within the One Health framework, it remains an ongoing challenge to attain a comprehensive understanding of the precise mechanisms governing AMR transmission. This includes the development of approaches for their determination and classification [175]. In this continually evolving landscape, the multifaceted dynamics of AMR underscore the need for further research and a multidisciplinary approach [170]. Elucidating the precise routes and mechanisms of AMR transmission is a pivotal step in the ongoing fight against this global public health threat, necessitating robust collaborative efforts among researchers, clinicians, and public health practitioners. In this context, the EPI-Net consortium, or “Epidemiology and Evolution of Pathogens Network”, represents a collaborative initiative aimed at advancing our understanding of the epidemiology and evolution of pathogens, especially in the context of infectious diseases. One of the primary objectives of EPI-Net is to facilitate the sharing of data, expertise, and resources among multiple research groups and institutions. The EPI-Net consortium plays a crucial role in advancing the understanding of antibiotic resistance within the context of genomic surveillance and One Health. The consortium recently developed a reporting guideline [176], with the objective of standardising data collection and reporting in the context of antimicrobial consumption and resistance surveillance within the One Health framework. This standardisation is vital for tracking the spread of antimicrobial resistance and evaluating the impact of interventions. Improving the quality and consistency of surveillance data, it enhances the ability to respond to and mitigate the growing threat of antimicrobial resistance.

## 8. Future Perspectives

The introduction of cutting-edge diagnostic tools stands as one of the most eagerly anticipated advancements in the management of antibiotic-resistant diseases [177]. This includes the prospect of direct sequencing from biological samples, particularly suited for infections with a high bacterial load, such as urinary tract infections, mastitis, or meningitis. Additionally, sequencing bacterial colonies derived from positive cultures demonstrates exceptional utility in the case of infections characterised by a low bacterial load, as seen in bloodstream infections [178]. These methodologies possess the potential to evolve into standard practices within the field of clinical microbiology.

The antibiotic resistome is continuously evolving, underscoring the need for advanced technologies to comprehensively understand its dynamics and diversity [179]. Furthermore, both the EUCAST and the ECDC have recognised that genome-based detection represents the future of AMR surveillance [96,180]. The utilisation of WGS technologies, for AMR detection and tracking confers the advantage of aligning with a One Health surveillance framework, facilitating precise comparisons across diverse reservoirs. As NGS technologies become increasingly accessible, metagenomics is poised to surge in popularity, thereby empowering clinicians to investigate the potential impact of various environments on resistance ecology [88].

## Figures and Tables

**Table 1 antibiotics-12-01580-t001:** Strengths and weaknesses of culture techniques and molecular methods. Abbreviations: PCR, polymerase chain reaction; RT, real time; NGS, next generation sequencing; MIC, minimum inhibitory concentration.

	Culture Testing	PCR/RT-PCR	Genomic Technologies
**Advantages**	Widely available and cheaper than NGS	Rapidly completed in 4–8 h	Pathogen identification and characterisation independent of culturing and microbe isolation
	Only basic equipment necessary	Increased identification of less common organisms such as viruses	Reduced time for slow growing pathogen identification, typing and characterisation
		Test multiple targets in one sample	Faster identification of outbreak clusters
		Higher sensitivity than culture testing	Higher sensitivity and specificity, better resolution
**Disadvantages**	Failures in the identification of fastidious bacteria and organisms that cannot be cultured	Higher costs than culturing	Higher costs than culturing
	Increased time for slow-growing pathogen identification and typing	High specific design of primers and need for a list of potential pathogens	Requirement of highly specialised laboratory equipment and trained personnel
	Requires special media and specific condition for the different microorganisms	Inability to determine the MIC of a compound	Inability to determine the MIC of a compound

**Table 2 antibiotics-12-01580-t002:** Comparison of second- and third-generation sequencing technologies. Abbreviations: WGS, whole genome sequencing.

	Second Generation—Short Reads	Third Generation—Long Reads
**Platforms**	Ion Torrent, Illumina	PacBio, Oxford Nanopore
**Sequencing principle**	Sequencing-By-Synthesis (SBS)	Single-Molecule Sequencing (SMS)
**Maximum reads length**	400 bp or 2 × 300 pb	>100 kb
**Advantages**	High sequence accuracy	Easier library preparation and portable technologies
	Able to sequence fragmented DNA	Suitable for the analysis of epigenetic markers
	High throughput (parallelisation of sequencing reaction)	Generation of very long reads
**Disadvantages**	Not able to resolve structural variants	Overall lower accuracy
	Not appropriate for analysing highly homologous genomic regions or epigenetic markers	Signals obtained from individual fragments can be weak
	Challenges for WGS due to the short reads	

**Table 3 antibiotics-12-01580-t003:** Comparison of the major genomic sequencing approaches.

	Methodology	Common Use
**Whole Genome Sequencing (WGS)**	Sequencing the entire genome	Identifying mutations, genomic structure
**Genome-Wide Association Studies (GWAS)**	Analysing genetic variations	Identifying associations between genetic variations and traits or diseases
**Target Sequencing (tNGS)**	Focusing on specific genomic regions	Studying specific genes or regions of interest
**Amplicon Sequencing**	Amplifying and sequencing specific DNA fragments	Microbial community analysis, genetic markers
**16S Metagenomics**	Sequencing the 16S rRNA gene	Studying microbial diversity and taxonomy
**Shotgun Metagenomics (mNGS)**	Sequencing all DNA in a sample	Analysing the entire genomic content of a microbial community

**Table 4 antibiotics-12-01580-t004:** Advantages and disadvantages of genomic methods for antimicrobial resistance detection. Abbreviations: MIC, Minimal Inhibitory Concentration.

	Advantages	Disadvantages
**Whole Genome Sequencing (WGS)**	Rapid results, especially for slow-growing pathogens	Complex data analysis
	Can predict drug resistance profile	Interdisciplinary collaboration needed for clinical integration
	Potential to unveil novel resistance mechanisms	Interlaboratory variability
	Accessible for other studies (e.g., phylogenetics)	Cost and resource barriers
	Supports public health decisions and policies	Regulatory and ethical concerns (data privacy and consent)
**Metagenomics** **(including amplicon and shotgun metagenomics)**	Faster and higher throughput than conventional methods.	Resource and time barriers to genome analysis
	mNGS is culture-independent and can identify all pathogens	Limitations in detecting unknown resistance genes or genetic variants
	Can detect and quantify numerous AMR genes without prior selection	Cannot provide MIC of a compound.
	Real-time AMR surveillance for outbreak detection.	
**Multiomics Analysis**	Provides insights into dynamic mechanisms behind AMR	Challenges in data integration, limited information for specific bacteria and resistance types
	Integration of genomics, transcriptomics, and proteomics offers a holistic view	Accounting for biological variations
	Improved accuracy of resistance forecasts.	Requires rigorous validation and interpretation

## Data Availability

No new data were created or analysed in this study. Data sharing is not applicable to this article.

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
