# Peer review of "Current Uses and Future Perspectives of Genomic Technologies in Clinical Microbiology"

_antibiotics, 2023, doi:10.3390/antibiotics12111580_

Round 1

Reviewer 1 Report

Comments and Suggestions for Authors

The manuscript is well-written, and the sections are well-structured. The manuscript serves as a review of molecular diagnosis of microorganisms and its contributions. The authors effectively list the technologies and make comparisons between them. Furthermore, they chronologically illustrate the evolution of this field. This manuscript is quite engaging as it provides a thorough comparison with classical microbiology techniques, highlighting their limitations and advantages.

Reviewer 2 Report

Comments and Suggestions for Authors

In the written manuscript authors explained the current use and future prospective of genomic technology in clinical microbiology.  

Reviewer comments are mentioned below:

The written manuscript should be revised carefully and arrange appropriate points in the appropriate section such as Line 346-362 written paragraph does not match the mentioned heading. It should be mentioned in AMR section in heading 3.

Abstract must be revised and rewritten again. In the abstract portion, sentences must be short and clear. Authors have written long sentences which have lack of clarity to reader such as Lines 8-10 and Lines 15-18 are long and not clear what authors want to say.

 In the manuscript, a brief introduction of review topic and what authors want to explore in the written manuscript should be mentioned in a separate heading as “Introduction”.

In the manuscript, major genomic sequence techniques which are generally in use should be summarized in a table format.

Line 25, 112, 180, 253, 280 and so on-Why capitalized each word of headings? Should change

Line 165-166: Sentences written in the metagenomics section is not completed and are unclear to reader. Must be revised  

Line 165 and Line 252: In some places such as line 165, multiple references are included in a single bracket but on the other side such as line 252 more than one reference has been included in separate bracket individually. Must be used single format to mention references in between text in entire manuscript.

Line 442: Authors should mention either manuscript or article in place of Paper.

Comments on the Quality of English Language

Minor English corrections and grammatical errors need to be checked and updated in the revised manuscript 

Reviewer 3 Report

Comments and Suggestions for Authors

This manuscript highlights recent advancements in sequencing technologies and data analytics, emphasizing their role in high-throughput pathogen detection and typing, which is becoming the standard for infectious disease surveillance and control. Genomic data are shown to enhance national surveillance systems and combat antimicrobial resistance, a global public health concern. Additionally, genome sequencing aids in hospital outbreak investigations, supports the One Health initiative's goal of holistic disease control, and, when combined with epidemiological data, predicts population trends and species transmissions, enhancing our understanding of antimicrobial resistance in various contexts. Here are some suggestions for this manuscript,

-       Section 1

  1. The publication provides a comprehensive overview of nucleic acid sequencing technologies and their evolution, which is highly informative. It would be beneficial to provide some information on the practical applications of each sequencing technology, especially in the context of biomedical research and diagnostics.

2.     The review could benefit from a discussion of the limitations and challenges associated with each sequencing method, as this would provide a more balanced perspective on their utility.

3.     The introduction mentions the increasing use of NGS in clinical microbiology laboratories, but it would be useful to elaborate on specific clinical applications and case studies to illustrate the impact of these technologies in healthcare settings.

  1. I would suggest that the authors consider providing recent examples or breakthroughs in the field of nucleic acid sequencing to highlight its ongoing significance in biomedical research and diagnostics.

-       Section 2

5.     The publication provides a valuable overview of different sequencing techniques, including Whole Genome Sequencing (WGS), Genome-Wide Association Studies (GWAS), Targeted Sequencing (tNGS), and Metagenomics (amplicon-based and shotgun/mNGS).

6.     The author mentions the potential of WGS for identifying antimicrobial resistance genes, but does not discuss the challenges associated with interpreting the data and translating it into clinical practice. A more in-depth discussion of this aspect would be beneficial for readers.

7.     The discussion of GWAS in microbiology is insightful, but it would be helpful to elaborate on specific challenges associated with bacterial GWAS and potential strategies for optimization.

8.     It would be beneficial to include recent examples or case studies showcasing the practical applications of these sequencing techniques in research or clinical settings.

-       Section 3

9.     In this section the authors provide a clear summary of the current state of genomic approaches to detecting antimicrobial resistance (AMR). While the article mentions the limitation of genomic methods in detecting chromosomal alterations that cause changes in gene expression, it would be helpful to expand on this point and discuss potential strategies for addressing this challenge. For instance, the authors could describe emerging technologies or computational methods that enable the detection of gene expression changes or discuss the potential integration of genomic data with other types of data (e.g., transcriptomic, proteomic) to improve the accuracy of AMR predictions.

-       Section 4

10. The description of "reverse genomic epidemiology" as a strategy for tracking bacterial transmissions and outbreaks is a novel and interesting concept that enriches the discussion. However, it would be beneficial to provide more details or examples of how this strategy is applied in practice.

11. The brief mention of k-mer typing is informative but could benefit from additional details or examples to help readers grasp the concept and potential applications more fully.

-       Section 5

12. Lane 302 – 311 This section correctly points out the need for expertise in both wet and dry labs for genomic surveillance. However, it could expand on the challenges and requirements in establishing and maintaining such capabilities, especially in resource-limited settings. Additionally, discussing the evolving technology landscape, such as the shift from MiSeq to MinION, is informative, but providing context on these platforms' pros and cons would enhance the discussion.

-- Section 6

13. This section effectively emphasizes the role of genomics within the One Health framework, and rightly stresses the importance of One Health genomic surveillance in controlling zoonoses and antimicrobial resistance (AMR). However, it might be beneficial to mention specific examples of how genomics has contributed to One Health initiatives or projects. it could delve deeper into how genomics can address these challenges and provide concrete examples of successful One Health genomic surveillance programs.

14. lane 381-391 It accurately mentions the necessity for collaboration between human health, animal health, agricultural sectors, and environmental monitoring. However, it might benefit from elaborating on the challenges and benefits of such collaborations and how genomics can facilitate communication among these sectors.

15.  Lane 401-403 The reference to metagenomics as a tool for detecting rare zoonotic transmissions is pertinent. However, it could offer more insight into the advantages and challenges of metagenomics in surveillance systems.

16. Lane 411-425 The discussion on AMR surveillance within the One Health context is crucial. However, it could go into more detail about the practical application of genomics in AMR surveillance, such as how it aids in understanding transmission routes and identifying antibiotic resistance genes. Providing concrete examples or case studies would make this section more informative.

Overall, this manuscript offers a forward-looking perspective on the pivotal role genomics can play in combatting antibiotic-resistant diseases, shedding light on the promising potential of cutting-edge diagnostic tools and surveillance methods on the horizon. However, there is room for improvement, both in terms of refining and expanding certain aspects to deepen its relevance. Incorporating additional real-world examples or case studies would significantly bolster its practical applicability and overall impact.

Reviewer 4 Report

Comments and Suggestions for Authors

This is an interesting review. I enjoyed the reading. I think overall the manuscript can benefit from some English revisions. Authors tend to have long sentences connected with and. I pointed out some of them.

-Line 8 through 9, is a running long sentence and has two ideas connected by “and”. Please separate it into two sentences and remove the “and” in line 9.

-Line 12, the abbreviation “AMR” was not explained in the abstract. Please explain the abbreviation before using it.

-Line 30, is a very long sentence. Please consider breaking it into two before the “NGS applications” and remove the “and”

-Table 1: “not culturable organisms” is not correct English. Please revise and correct.

-None of the abbreviations in the table is explained “MIC, NGS, PCR, RT-PCR”

-Table 2: “WGS” was not explained

-In the second section “the second generations sequencing”

Please add more discussion on Ion Torrent by ThermoFischer. The platform generates full length reads and is widely used in different applications.

-The section of “status and perspectives” is very well written. I would suggest summarizing the pros and cons in a table.

-Line 257, Please consider make the sentence starting with “these methods” a new sentence and correct punctuation accordingly.

-Line 262, this is a long running sentence. Please break it into two and start the next sentence at “Nevertheless”

-Please revise the sentence lines 294 through 297. Two complete sentences are incorrectly and unnecessarily connected. Please consider separating at the word “rapid”

-Line 300, this should be a new sentence. Please capitalize “However” and use the appropriate punctuation

Line 301, please change the word “and” into “or”, it is grammatically more correct.

Line 324, please consider make the sentence starting with “consequently” a new sentence and correct punctuation accordingly.

Line 328, please correct “outmost” to “utmost”

Line 328, please start a new sentence with “although” and use the appropriate punctuation.

Line 336, please start a new sentence with “In a typical year” and use the appropriate punctuation

Line 337, please revise and correct the English in the connecting sentence starting with “from which”.

Line 339, please start a new sentence with “significant differences” and use the appropriate punctuation.

Line 412, please separate these two sentences and remove the “and”.

Line 417, please separate these two sentences after the reference “112”

Line 439, please start a new sentence with “Metagenomics” and use the appropriate punctuation.

Comments on the Quality of English Language

I think overall the manuscript can benefit from some English revisions. Authors tend to have long sentences connected with and. I pointed out some of them.
